# Effects of decision-making on indoor bouldering performances: A multi-experimental study approach

**Jerry Prosper Medernach** [ORCID] *, **Daniel Memmert**

Institute of Exercise Training and Sports Informatics, German Sport University Cologne, Cologne, Germany

* j.medernach@lb.dshs-koeln.de

## Abstract

The purpose of this study was to investigate whether novice, intermediate, and advanced bouldering athletes would differ in their decision-making abilities and to what extent distinct problem-solving tactics would affect the athletes' bouldering performances. Seventy-seven male bouldering athletes participated in a multi-experimental study with the conceptual replication of three bouldering tasks. Participants were allocated according to their ability levels to the novice group (NOV with $n = 18$), the intermediate group (INT with $n = 18$), or the advanced group (ADV with $n = 41$). The data collected for movement analysis via video consisted of the pre-ascent decision-making times, the number of movement deviations from the best solution, the number of movement mistakes, the average gripping times, the bouldering times to the top, the number of successful ascents, and the number of attempts to complete the tasks. Results among all three experiments revealed shorter decision-making times, fewer movement mistakes, and shorter average gripping and bouldering times to the top in the ADV group than in the NOV group and the INT group. Furthermore, participants from the ADV group demonstrated fewer movement deviations than participants from the NOV group (in all three experiments) and the INT group (Experiment 1 and Experiment 2). Moreover, participants from the ADV group and the INT group were characterized, in all three experiments, by a higher number of successful ascents and a lower number of attempts to complete the tasks than participants from the NOV group. In total, these findings emphasize that accomplished decision-making abilities consist of a key determinant in successful indoor bouldering performances.

## Introduction

Competitive bouldering was to be included in the Tokyo 2020 Olympic Games, which emphasizes that bouldering is increasingly popular and competitive [1]. Although the emergence of climbing and bouldering as competitive and recreational activities has sparked research interest (e.g., recovery methods [2]; injury risk factors [3]; movement behaviour [4]; grip strength and endurance [5]), no empirical research has been conducted in regard to investigating the effects of decision-making on indoor bouldering performances.

**Data Availability Statement:** All relevant data are within the manuscript and its Supporting information files. Further files are available from: https://doi.org/10.7910/DVN/SUJMT7.

**Funding:** The authors received no specific funding for this work.

**Competing interests:** The authors have declared that no competing interests exist.

Decision-making in sports can be understood as the cognitive process of selecting a response to a stimulus from a range of available options in circumstances where an action is needed [6,7]. According to Baker, Côté, and Abernethy [8], decision-making in team sports consists of the ability to perceive essential information from the playing environment, to correctly interpret this information, and to select the appropriate response. In the ecological dynamics approach, decision-making is understood as an emergent behaviour embedded in a performer-environment relationship with the intentional exploration for efficient solutions [9]. Recent studies emphasize the pivotal role of decision-making in sports, particularly in open, fast-paced, and dynamic sport activities [6]. Kempe and Memmert [10], for instance, examined the decision-making skills of elite soccer players and observed more creative decisions, contributing to a more successful completion of the last eight game actions before the actual shot on the goal among teams that advanced into the later rounds of the tournament than in less successful teams. Furley and Dörr [11] examined the decision-making abilities of surfers and observed that the athletes' decisions corresponded to a linear manner with their amount of surfing experience. More precisely, the authors found that experienced surfers could better distinguish between surfable and non-surfable waves compared to less experienced or non-surfing controls. Furthermore, Connors, Burns, and Campitelli [12] investigated the role of search behaviour in chess by replicating the study of de Groot (1946) and found that experienced chess players were characterized by higher pattern recognition and better decision-making abilities than less experienced players.

Indoor bouldering is a climbing discipline undertaken without ropes on approximately four-metre-high artificial walls with landing mats to ensure safety [13–15]. During international bouldering competitions, athletes must attempt four to five bouldering tasks that involve short (i.e., an average of four to eight handholds) but strenuous, complex, and coordinative climbing movements [14,15]. Bouldering competitors are generally ranked by their bouldering scores, which consist of both the total number of completed tasks (i.e., number of tops) and the number of attempts they require to complete each task [15]. To achieve high bouldering scores, competitors must, therefore, strive to promptly identify suitable ascent-tactics and make appropriate decisions to complete each task in as few attempts as possible and to circumvent expendable movement mistakes (i.e., inappropriate or erroneous movements relative to the movement demands) that may result in the failure to complete a task [4,16].

In international bouldering competitions, major constraints on generating appropriate problem-solving tactics consist of the prohibition to physically rehearse the tasks and the time limitation athletes are confronted with [15]. Medernach, Kleinöder, and Lötzerich [4] observed that boulder world cup competitors perform, in general, multiple attempts to complete a task (i.e., on average 4–5 attempts) with relatively long bouldering times (i.e., on average 30–40 s), relatively short rest times between the attempts (i.e., approximately 30 s), and limited recovery times between two consecutive tasks (i.e., five minutes in the qualification and semi-final rounds). Furthermore, bouldering athletes must, despite standardized framework conditions (e.g., no direct opponents, weather conditions) and a limited number of fundamental movement patterns (e.g., back step), constantly adapt their problem-solving tactics to the specific movement demands of the bouldering tasks [16,17]. In the open-closed continuum for motor skills originally presented by Poulton (1957), indoor bouldering can, therefore, be understood as a relatively open-skill activity. Specifically, athletes are confronted with a considerable variability of movement demands [17–19] due to an almost infinite number of climbing hold arrangements (e.g., sizes, shapes, orientation) and constant modifications of the wall features (e.g., inclination, volumes).

When following the cognitive decision-making model by Memmert [20], the first cognitive subprocess in decision-making consists of anticipating the task (i.e., stimulus) athletes are

confronted with (Fig 1). In the following cognitive subprocesses, athletes perceive the stimulus through a bouldering preview, pay attention to the specific movement demands, and compare them with previous experiences (i.e., movement repertoire). In the last cognitive subprocesses, athletes generate potential problem-solving tactics and choose the most appropriate and thus the best solution from the previously generated problem-solving options. The best solution is understood as the most straightforward option to climb a task [16,18] and consists of the ascent-tactic that 'best fits' the task, relative to the athletes' physical constraints and motor action capabilities [17]. Consequently, the best solution can vary on personal characteristics (e.g., body height) and individual motor skills (e.g., finger strength), which athletes must integrate into the cognitive decision-making processes to generate efficient motor sequence plans [16,17].

The model of cognitive decision-making in indoor bouldering (Fig 1) emphasizes, in agreement with Sanchez, Torregrossa, Woodman, Jones, and Llewellyn [16], that generating suitable problem-solving tactics in indoor bouldering is related to the athlete's movement repertoire (i.e., movement knowledge based on previous experiences and deliberate practice) and bouldering preview skills. In fact, athletes must integrate their movement repertoire into the environmental information previously acquired through the bouldering preview, in order to accurately interpret the movement demands and to facilitate the selection of suitable responses to the task [11,16]. More precisely, a bouldering preview consists of the pre-ascent visual inspection of the bouldering task, which is commonly applied by athletes to pick up visual cues of the task and perceive potential affordances. Following Gibson [21], affordances can be understood as action possibilities (e.g., positively shaped hand holds that are unchallenging to grasp) provided by the environment (i.e., bouldering setup) and perceived by the athletes, relative to their physical constraints and motor skills (i.e., interaction between the environment and the observer). Perceiving affordances thus consists of an initial cognitive subprocess while visually inspecting and interpreting the movement demands (i.e., perception of the stimulus). The final decision, however, as to which affordances athletes integrate into their problem-solving tactics and motor plans depends, in particular, on the movement demands of the bouldering task and the athletes' physical constraints and motor skills [9]. Pezzulo, Barca, Bocconi, and Borghi [19] observed that experienced climbers demonstrated a better recall of climbing hold positions following a route preview than novice climbers. Interestingly, this effect was only present on a difficult climbing route in line with the experts' ability levels, but there was no difference in the recall abilities on both an easier and impossible route. The authors argued that the mental simulation of movement sequences and the perception of affordances are related to the athlete's motor expertise. This means that athletes who lack motor skills for solving a particular climbing task risk being impeded from the correct simulation of the movement sequences and cannot generate suitable problem-solving tactics [19]. Similar findings were recently reported by Whitaker, Pointon, Tarampi, and Rand [17] who observed that climbing expertise was positively associated with the perceptual judgement accuracy of action capabilities (i.e., ability to determine whether a single climbing movement could be performed) and visual memory abilities (i.e., memorization of climbing holds). The findings of Whitaker et al. [17] are consistent with the results of Boschker, Bakker, and Michaels [22] who observed that expert climbers focused during the route preview more on functional aspects of the climbing task and recalled, therefore, more information following a route preview compared to less experienced controls.

In addition, research in the domain of sport climbing emphasizes that motion fluidity (i.e., how fluid athletes perform a movement sequence) and climbing velocity (i.e., ascent pace) appropriate to the movement demands (e.g., difficulty degree, resting points) can be considered salient determinants of the athlete's problem-solving accuracy [4,18,23]. Rodio, Fattorini,

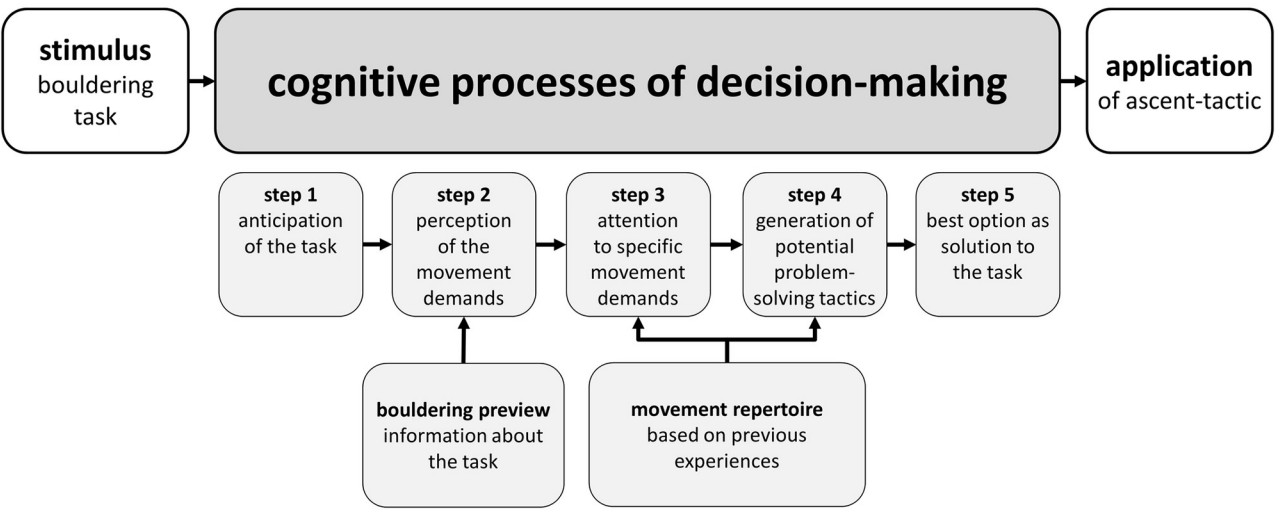

**Fig 1. The model of cognitive decision-making processes in indoor bouldering (retrieved and adapted from Memmert [20]).**

Rosponi, Quattrini, and Marchetti [24] found in experienced athletes a higher climbing velocity and lower energy expenditure than in less experienced controls when ascending the same climbing task. Similar effects were reported by España-Romero et al. [25] who observed that repeating the same climbing route contributed to a reduction in the athletes' ascent-times and absolute energy expenditure. Specifically, the authors argued that more economic movement actions through repeating the same task could contribute to higher motion fluidity and, therefore, to a reduction of the overall isometric work. In a further study, Sanchez, Lambert, Jones, and Llewellyn [26] found that route previewing did not influence the athletes' climbing performances (i.e., whether participants could complete the task) but resulted in fewer and shorter non-movement times while ascending the route. Altogether, these findings emphasize that more suitable ascent-tactics, through route previewing or repeating the same task, can contribute to shorter non-movement periods and thus to lower ascent-durations, which in turn, can hinder early local muscular fatigue [4,13,16,23].

The purpose of the present research program was to investigate the effects of decision-making on indoor bouldering performances. To be precise, our study aimed to examine whether novice, intermediate, and advanced bouldering athletes would differ in their decision-making abilities and to what extent distinct problem-solving tactics would affect the athletes' bouldering performances. We established the following hypotheses: Firstly, we hypothesized, in extension to Sanchez et al. [16], that accomplished bouldering athletes would require shorter pre-ascent decision-making times to generate their problem-solving tactics than less-experienced athletes. Secondly, we hypothesized, by following Sanchez et al. [16] and Whitaker et al. [17], that accomplished bouldering athletes would be characterized by higher decision-making abilities and more suitable problem-solving tactics than less-experienced controls. In extension to Sanchez et al. [16] and España-Romero et al. [25], we thirdly hypothesized that more appropriate problem-solving tactics among accomplished bouldering athletes would also contribute to shorter average gripping and bouldering times to complete a task.

By following Simons [27] and Lynch, Bradlow, Huber, and Lehmann [28], we implemented a multi-experimental study design with the conceptual replication of three bouldering tasks to test our hypotheses on athletes with different ability levels and under various, ecologically valid experimental conditions. In Experiment 1, we examined the decision-making abilities and

problem-solving tactics of novice, intermediate, and advanced bouldering athletes on a novel bouldering task set in accordance with the IFSC competition rules (i.e., International Federation of Sport Climbing). In Experiment 2, we replicated the bouldering task of Experiment 1 by implementing slight handhold modifications. These handhold modifications increased the opportunity for misperceiving affordances and contributed, therefore, to a higher decision-making complexity without increasing the difficulty degree or distinctly modifying the movement demands compared to the previous Experiment 1. In Experiment 3, we further increased the complexity of decision-making by implementing a bouldering task with many climbing holds varying in size, shape, and colour. Although the climbing hold arrangement permitted multiple ascent-solutions, most handholds were hard to grasp and participants thus had to accurately visualize during the bouldering preview the well-shaped handholds to generate suitable ascent-tactics.

## Materials and methods

### Participants

Eighty-six bouldering athletes volunteered to participate in the three experiments. Athletes were randomly recruited (i.e., they were arbitrarily addressed) from local climbing associations and commercial climbing centres (period of recruitment: November 2019) with the purpose to provide a high number of participants. By following Medernach et al. [14], participants had to be at least 18 years old, healthy, and with no recent injuries. Furthermore, a minimum of two-week bouldering experience was prescribed to ensure fundamental bouldering skills. Since only nine out of eighty-six subjects were female, we decided to exclude the female athletes from the study. This contributed to a higher body characteristics consistency (e.g., body height) among the 77 male participants that were retained for data analysis. Participants were informed verbally and in writing about the purpose, the contents, and the duration of the investigation. They had to fill out a declaration of consent and were informed of their right to leave the study at any stage. The study protocol was in accordance with the Code of Ethics of the World Medical Association (Declaration of Helsinki) and had ethical approval from the ethics committee of the German Sport University.

Personal characteristics, previous sport-specific experiences, and the current bouldering ability levels of the participants are displayed in Table 1. The self-indication of the bouldering ability level by using the Fb grading scale (i.e., Fontainebleau scale with an open-ended numerical system ranging from 1 to 9; the letters A, B, and C are used as suffixes, and the plus and minus signs indicate further difficulty differences) consists of a valid and reliable assessment method [29]. It was applied to allocate participants to the novice group (NOV with $n = 18$; 1–4 FB), the intermediate group (INT with $n = 18$; 5-6B+ FB), or the advanced group (ADV with $n = 41$; 6C-7B+ FB). By following Draper et al. [29] and Brent, Draper, Hodgson, and Blackwell [30], reports from the Fb scale were converted into a numerical benchmark (i.e., study score; NOV: 1–2; INT: 3–7; ADV: 8–10) to enable statistical analysis. In extension to Draper et al. [29], the self-perception of technical skills was assessed as an additional parameter by using a 5-point Likert scale (level 1: novice technical skills; level 2: intermediate technical skills; level 3: advanced technical skills; level 4: elite technical skills; level 5: world-class technical skills). The intra-class correlation coefficient ($r = 0.819$; $p < 0.001$) indicated a high correlation between the perceived technical skills and the bouldering ability levels of the athletes.

### Investigation procedure

In accordance with Medernach, Kleinöder, and Lötzerich [14], participants were instructed in the interview prior to the investigation to respect a minimum rest period of 48 hours (i.e.,

**Table 1. Subject characteristics of the novice group (NOV), the intermediate group (INT), and the advanced group (ADV).**

| | NOV (*n* = 18) | INT (*n* = 18) | ADV (*n* = 41) |
|---|---|---|---|
| age (*years*) | 24.5 ± 5; *p* = 0.582 | 27.1 ± 6; *p* = 1.00 | 27.3 ± 7; *p* = 0.291 |
| | $F_{(2,76)} = 1.49$, *p* = 0.232 | | |
| height (*cm*) | 177.2 ± 4; *p* = 1.00 | 178.8 ± 6; *p* = 1.00 | 179.8 ± 6; *p* = 0.256 |
| | $F_{(2,76)} = 1.53$, *p* = 0.224 | | |
| weight (*kg*) | 72.4 ± 7; *p* = 1.00 | 71.3 ± 7; *p* = 0.481 | 68.7 ± 6; *p* = 0.150 |
| | $F_{(2,76)} = 2.35$, *p* = 0.103 | | |
| sport-specific experiences (*years*) | 0.4 ± 0.1; *p* = 0.929 | 1.6 ± 1; *p* < 0.001 | 5.1 ± 2; *p* < 0.001 |
| | $F_{(2,76)} = 13.45$, *p* < 0.001, *r* = 0.5 | | |
| competition experiences (*years*) | / | / | 3.4 ± 2 |
| bouldering ability levels (*study score*) | 2.7 ± 1; *p* < 0.001 | 6.0 ± 1; *p* < 0.001 | 9.9 ± 1; *p* < 0.001 |
| | $F_{(2,76)} = 60.5$, *p* < 0.001, *r* = 0.9 | | |
| bouldering ability levels (*Fb*) | 4 | 6B | 7A+ |
| bouldering ability groups (*classification*) | *novice* | *intermediate* | *advanced* |
| self-perceived technical skills (*score*) | 1.4 ± 0.5; *p* < 0.001 | 2.7 ± 0.6; *p* < 0.001 | 3.9 ± 0.8; *p* < 0.001 |
| | $H_{(2)} = 151.5$, *p* < 0.001, *r* = 0.5 | | |
| grip strength (*N*) | 285.8 ± 51; *p* = 0.740 | 298.9 ± 107; *p* < 0.001 | 468.5 ± 95; *p* < 0.001 |
| | $F_{(2,76)} = 36.52$, *p* < 0.001, *r* = 0.7 | | |

Results are given as mean ± standard deviation except for the ability levels and the ability groups. An alpha level of *p* < 0.05 was used to determine statistical significance. The ability levels are indicated as Fb-values (i.e., Fontainebleau grading scale). The climbing ability conversion tables by Draper et al. [29] and Brent et al. [30] were used to determine the study scores (novice: 1–2; intermediate: 3–7; advanced: 8–10) and the ability classification (novice: 1–4 Fb; intermediate: 5-6B+ Fb; advanced: 6C-7B+ Fb).

although this was not explicitly examined), during which no unnecessary physical activity was permitted. All three experiments (period of exposure and data collection: January to February 2020) took place on the same day with standardized recovery times of 20 minutes between each experiment. As during bouldering competitions, participants had to remain before each experiment in an isolation zone to guarantee standardized observation procedures. They were then given, in each experiment, an individual warm-up period of a standardized 15-minute duration, followed by a rest time of four minutes, during which (i.e., only in Experiment 1) body characteristics and grip strength were assessed. To guarantee optimal individual physiological and psychological preparations, a standardized warm-up protocol was not implemented (in accordance with Sanchez et al. [26]). Participants were then introduced to the investigation procedure and the IFSC competition regulations. An attempt at the bouldering task was defined in accordance with the IFSC rules as starting at the point when a participant leaves the ground and has both hands on the marked handholds [15]. The successful completion of the task required achieving a controlled position at the marked finishing hold with both hands [15]. Participants were instructed to ascend the bouldering tasks with a minimum number of attempts and by applying an individual bouldering pace. Furthermore, they were instructed to climb as fluently as possible by minimizing non-movement actions and to seek the most appropriate problem-solving path for each task. Prior to the first experiment, participants were given a familiarization trial on a separate bouldering task (excluded from data collection), followed by a standardized rest time of four minutes.

In a further step, participants were given, in each experiment, a standardized decision-making time of four minutes to visually inspect the bouldering tasks and to generate pre-ascent problem-solving tactics. Subjects were not informed about the difficulty degrees and were not

allowed to physically rehearse the tasks. As in bouldering competitions, participants received information about only the starting and finishing positions with no additional information concerning the movement demands, grip sizes, or potential problem-solving tactics. Once the participants felt confident in their sequences or the four-minute decision-making time was up, they were asked to turn their back to the bouldering wall to prevent a further visual inspection. By following the IFSC rules, participants were then given, for each experiment, a standardized ascent-time of four minutes. A timing system was used to display the remaining decision-making and bouldering time.

### Data collection procedure

**Anthropometrics and strength.** Subjects were weighed in shorts and t-shirts without shoes to the nearest 0.1 kg using a Seca 760 scale (Seca GmbH, Hamburg, Germany) and height was measured without shoes to the nearest 0.5 cm using a Seca 213 stadiometer (Seca GmbH). Although finger strength was unlikely to have an impact on whether athletes would follow the best solution (i.e., see design of the bouldering task sections for further details), handheld grip strength, which consists of a valid and reliable method for assessing sport-specific finger strength [31], was determined using a calibrated Smedley Spring dynamometer (Saehan; Gyeonggi-do, KR) to provide additional information about the participants' physical capabilities. The maximum pressure applied without the use of the thumb to enable a sport-specific test implementation and the grip span was adjusted to reach the phalanx distalis of the ring finger [14]. Subjects had to perform three attempts with the dominant hand, gradually applying maximal pressure for two seconds. The highest score of three attempts was recorded with a standardized rest period of one minute between efforts.

**Movement and decision-making analyses.** A Sony FDR-AXIEB 4K Ultra-HD-Camcorder (Sony Corporation, Minato, Tokyo, Japan) was used for video recording. Video analyses to examine the effects of decision-making on indoor bouldering performances were performed using Adobe Premiere Pro CS6 software (Adobe Systems Corporation, San José, California, USA) and consisted, in accordance with Medernach et al. [4], of the following items (independent variables): (a) the pre-ascent decision-making times, (b) the number of movement deviations from the best solution, (c) the number of movement mistakes performed by the participants, (d) the average gripping times, (e) the bouldering times to the top, (f) the number of tops, and (g) the number of attempts to complete the task.

The pre-ascent decision-making time was limited to four minutes and described the participants' duration to visually inspect and interpret the movement demands trough a bouldering preview. The decision-making time can be considered a determinant of how promptly athletes decipher the corresponding movement demands and how rapidly they generate problem-solving tactics relative to the task. Shorter decision-making times, therefore, reflect a faster perception of visual cues and potential affordances [4,16,17]. The number of movement deviations was defined as the total number of hand and foot movements that deviated from the expert route setters' best solution. Consequently, movement deviations determine how accurately the athletes' applied ascent-tactics followed the best solutions of the tasks as perceived by the experts. A movement mistake was defined as the failure to grasp a target handhold and thus to successfully complete a particular movement due to inappropriate or erroneous hand or feet actions. This means that movement mistakes resulted either in the athletes' falling off the wall or the athletes' obligation to adapt their initial ascent-tactics (e.g., raise the foot) to grasp the target handholds. A low number of movement mistakes emphasizes a more appropriate interpretation of the movement demands and thus a better decision-making ability, contributing to a more suitable problem-solving tactic that requires fewer ad hoc adaptations of the initial

ascent-tactic while attempting the task. The average gripping time was calculated as the mean of each period from the grasping of a handhold to its release and the grasping of the next hand-hold. The bouldering time to the top was defined as the duration required by the athletes to successfully complete the task by the IFSC rules. Shorter average gripping and bouldering times to the top are determinants of the athlete's pre-ascent problem-solving accuracy as appropriate ascent-tactics contribute to higher motion fluidity and shorter non-movement times, which are generally required by athletes to interpret the movement demands while ascending a task [19]. The number of tops was defined as the number of participants who could complete the bouldering tasks by the IFSC rules. Both, the number of tops and the number of attempts to complete the tasks were used to determine the participants' bouldering scores in accordance with the IFSC rules. High bouldering scores emphasize that athletes could promptly identify suitable ascent-tactics, contributing to fewer attempts that are necessary to solve a task [4,16]. In addition to the seven independent variables, video analyses included in the Experiment 2 the assessment of the number of error grips (i.e., handholds that were impossible to grasp), and in the Experiment 3 the assessment of the number of key grips (i.e., handholds that were unchallenging to grasp) and the total number of handholds used by the participants to ascent the bouldering task.

Two scientific experts with sport-specific qualifications and elite bouldering ability levels performed the video analyses. Each expert independently determined the independent variables and the intra-class correlation coefficient (ICC) was used to assess consistency between the experts, with (a) $r = 0.920$ for the pre-ascent decision-making times, (b) $r = 0.895$ for the movement deviations from the best solution, (c) $r = 0.925$ for the movement mistakes, (d) $r = 0.910$ for the average gripping times, (e) $r = 0.995$ for the bouldering times to the top, (f) $r = 1$ for the number of tops, and (g) $r = 1$ for the number of attempts.

## Statistical analyses

Statistical analyses were performed using IBM SPSS Statistics 26 (IBM Corporation, Armonk, NY, USA) and Microsoft Excel 2019 (Microsoft Corporation, Redmond, WA, USA). Data are reported as the mean values and standard deviations and an alpha level of $p < 0.05$ (2-tailed) was used to determine statistical significance. The 95% confidence interval of the mean (95% CIs) is indicated for interval-scaled variables. All variables were assessed for their normality of distribution using (a) a one-sample Kolmogorov-Smirnov test and (b) a skewness and kurtosis $z$ value test. One-way independent analysis of variance (ANOVA) was used to determine group differences. The non-parametric Kruskal-Wallis one-way analysis of variance was implemented when the assumptions for ANOVA were violated. A multivariate analysis of variance (MANOVA) was implemented to examine participants' movement and decision-making abilities. A priori power analysis with $f^2(V) = 0.1$, $\alpha = 0.05$, power $(1-\beta) = 0.8$, and seven response variables (i.e., independent variables) indicated a total sample size of $n = 81$, which approximately corresponds to the 77 participants in the present research program. Levene's test was used to verify the homogeneity of variance and Box's test was used to investigate the homogeneity of covariance matrices. Post-hoc statistical power was found to be 0.9 or higher in ANOVA and 1.00 in MANOVA. Bonferroni's and Hochberg's GT2 post-hoc comparisons were used to determine differences between the ability groups. The square root of eta-square was used to determine the effect sizes. Comparisons within the groups were performed by using the non-parametric Mann-Whitney test with Cohen's $d$ used to describe the effect sizes. The non-parametric Wilcoxon test was used for comparisons between the experiments. Spearman's rank-order correlation was used to examine a significant relationship between two variables.

## Experiment 1

In Experiment 1, we investigated the decision-making abilities and problem-solving tactics of novice, intermediate, and advanced bouldering athletes while attempting a novel bouldering task. To be precise, we were curious about whether athletes with different ability levels would differ in their decision-making skills and to what extent distinct problem-solving tactics would affect their bouldering performances.

### Design of the bouldering task and assessment of the best solution

Three expert route setters (30.7 ± 3 yrs., 179.0 ± 4 cm, 66.5 ± 9 kg) were charged to set the task in Experiment 1. Expert 1 was a professional bouldering athlete with a world-class bouldering ability level (8b+ Fb; study score: 14), long sport-specific experiences (9 yrs.), and long route-setter experiences (6 yrs.). Expert 2 was a professional and qualified route setter with an elite bouldering ability level (8a Fb; study score: 12) and long sport-specific experiences (12 yrs.). Expert 3 was a scientific expert in the research field with an elite bouldering ability level (8a Fb; study score: 12), long sport-specific experiences (20 yrs.), and sport-specific coaching qualifications. The bouldering task was set on a 3.5 m high bouldering wall and included, in accordance with the IFSC rules, a total of eight handholds (Fig 2). The difficulty degree of the task was set at 5B Fb and was thus distinctly below the average bouldering ability level of the subjects. Furthermore, experts used positively shaped climbing holds that were unchallenging to grasp and to step on with relatively low distance between the handholds. The design and the difficulty degree of the task entailed relatively simple and undemanding movements to ensure that the task was technically and physically climbable by all the participants and to minimize the impact of interfering factors such as physical constraints (e.g., body height) or motor capabilities (e.g., grip strength, upper body power) on the participants' decision-making performances.

Following the route setting, experts independently determined the most straightforward ascent-path and thus how they perceived to best climb the task. For this purpose, each expert established a synoptic best solution movement pattern consisting of the successive

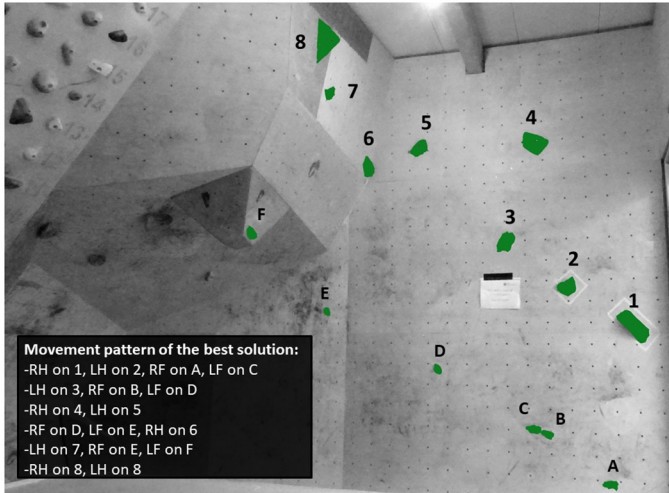

**Fig 2. The bouldering task in Experiment 1 (RH: Right hand; LH: Left hand; RF: Right foot; LF: Left foot).** Two handholds marked the starting position (i.e., handhold number 1 and 2) and the last hold (i.e., handhold number 8) marked the top. The experts' best solution is displayed in the movement pattern.

displacement of hand and feet movements. The synoptic movement patterns were converted into numerical codification to enable statistical analysis. The intra-class correlation coefficient (ICC) was used to assess consistency between the experts and indicated perfect interrater reliability, with $r = 1$. Consequently, all experts agreed on the best solution and confirmed that potential movement deviations from the best solution (e.g., grasping the handholds number three and six with both hands) were distinctly more strenuous and less straightforward. In order to additionally assess the reliability and consistency of the best solution as perceived by the experts, 13 female bouldering athletes (27.5 ± 3 yrs., 166.1 ± 8 cm, 52.8 ± 6 kg) with elite bouldering ability levels (7C Fb; study score: 10.6 ± 1) volunteered to participate in a retest procedure. The retest procedure aimed to investigate whether differences in physical constraints or individual capabilities would affect the athletes' ascent-tactics. Results indicated non-significant differences in the ascent-tactics ($U = 6.000$; $p = 0.350$; $r = 0.1$) between the experts and the female bouldering athletes despite significantly lower body height ($U = 1.000$; $p = 0.020$; $r = 0.6$) and absolute grip strength results ($U = .000$; $p < 0.001$; $r = 0.9$) among female athletes. These findings confirm the consistency of the ascent-tactic between the experts and the female bouldering athletes despite distinct physical constraints (i.e., body height) and individual capabilities (i.e., grip strength).

## Results

ANOVA testing revealed non-significant effects between the ability groups for age, body height, and body weight. In contrast, significant differences between the groups were found in the sport-specific experiences, the bouldering ability levels, the self-perceived technical skills, and grip strength (see Table 1). MANOVA revealed significant differences in the movement and decision-making abilities between the groups (see Table 2), with Pillai's trace $V = 1.02$, $F (14,132) = 9.7$, $p < 0.001$, $r = 0.9$, SP = 1.00. In fact, the pre-ascent decision-making times, the number of movement deviations and movement mistakes, and the average gripping and bouldering times to the top were significantly lower in the ADV group than in the NOV and INT groups. Relatedly, Spearman's rho indicated a significant negative correlation between the bouldering ability levels of the participants and both the number of movement deviations ($r = -0.653$; $p < 0.001$) and the number of movement mistakes ($r = -0.746$; $p < 0.001$). Furthermore, results showed a significantly lower number of attempts required to complete the task in the ADV group than in the INT and NOV groups. In contrast, the number of tops did not differ between the ADV group and the INT group with, however, a significantly lower number of tops in the NOV group than in the INT and ADV groups.

## Discussion

The purpose of Experiment 1 was to examine whether novice, intermediate, and advanced bouldering athletes would differ in their decision-making abilities and, to what extent, distinct problem-solving tactics would affect the athletes' bouldering performances while attempting a novel bouldering task. Our results revealed that participants from the ADV group demonstrated higher bouldering scores, required shorter pre-ascent decision-making times, performed a lower number of movement mistakes and movement deviations from the best solution, and demonstrated shorter average gripping and bouldering times to the top than the less-experienced participants from the NOV and INT groups. In total, our findings emphasize that accomplished decision-making skills and suitable problem-solving tactics can be considered a key determinant in indoor bouldering.

The shorter pre-ascent decision-making times in the ADV group are related, in agreement with Sanchez et al. [16], to higher bouldering ability levels and longer sport-specific

**Table 2. Study results of the novice group (NOV), the intermediate group (INT), and the advanced group (ADV) in Experiment 1.**

| | NOV ($n$ = 18) | INT ($n$ = 18) | ADV ($n$ = 41) |
|---|---|---|---|
| decision-making times ($s$) | 94.3 ± 33 [78–111] | 47.7 ± 26 [37–61] | 33.5 ± 12 [29–36] |
| | $p < 0.001$; $d = 1.6$ | $p < 0.001$; $d = 0.8$ | $p < 0.001$; $d = 2.9$ |
| | | $F(2,76) = 45.92$, $p < 0.001$, $r = 0.7$ | |
| deviations from the best solution ($n$) | 3.4 ± 1.0 [2.9–4.0] | 2.3 ± 1.3 [1.7–3.0] | 1.0 ± 1.3 [0.6–1.4] |
| | $p = 0.027$; $d = 0.9$ | $p = 0.001$; $d = 1.0$ | $p < 0.001$; $d = 1.9$ |
| | | $F(2,75) = 24.87$, $p < 0.001$; $r = 0.6$ | |
| movement mistakes ($n$) | 3.8 ± 1.0 [3.4–4.5] | 2.2 ± 1.3 [1.5–2.9] | 0.7 ± 1.1 [0.5–1.3] |
| | $p < 0.001$; $d = 1.4$ | $p = 0.001$; $d = 1.3$ | $p < 0.001$; $d = 2.9$ |
| | | $F(2,75) = 47.65$, $p < 0.001$, $r = 0.8$ | |
| average gripping times ($s$) | 5.6 ± 1.4 [4.8–6.3] | 4.3 ± 0.9 [3.9–4.7] | 3.1 ± 0.7 [2.9–3.4] |
| | $p < 0.001$; $d = 1.1$ | $p < 0.001$; $d = 1.5$ | $p < 0.001$; $d = 2.6$ |
| | | $F(2,75) = 43.41$, $p < 0.001$, $r = 0.7$ | |
| bouldering times to the top ($s$) | 48.6 ± 13 [42–55] | 33.3 ± 11 [28–39] | 21.7 ± 7 [19–24] |
| | $p < 0.001$; $d = 1.3$ | $p < 0.001$; $d = 1.4$ | $p < 0.001$; $d = 2.9$ |
| | | $F(2,71) = 46.26$, $p < 0.001$, $r = 0.7$ | |
| number of tops (% / $n$) | 72.2 / 13; $p < 0.001$ | 100 / 18; $p = 1.00$ | 100 / 41; $p < 0.001$ |
| | | $F(2,71) = 22.68$, $p < 0.001$, $r = 0.6$ | |
| number of attempts ($n$) | 4.1 ± 1.8 [3.2–4.9] | 1.8 ± 0.8 [1.4–2.2] | 1.0 ± 0.2 [0.9–1.1] |
| | $p < 0.001$; $d = 1.6$ | $p = 0.027$; $d = 1.2$ | $p < 0.001$; $d = 3.0$ |
| | | $F(2,75) = 65.30$, $p < 0.001$, $r = 0.8$ | |

Results are given as mean (number or seconds) ± standard deviation with the 95% CI, except for the number of tops (percent and number). An alpha level of $p < 0.05$ was used to determine statistical significance.

experiences among more experienced participants. Specifically, due to longer sport-specific experiences, participants from the ADV group could benefit from a higher movement knowledge and better motor expertise while generating their problem-solving tactics. The relatively low difficulty and the unchallenging movement demands of the task permitted athletes from the ADV group to promptly decipher the movement demands and to rapidly generate their problem-solving tactics. In contrast, video analyses revealed that participants from the NOV group required approximately three times longer to generate their ascent-tactics than participants from the ADV group. Our findings highlight, in agreement with Pezzulo et al. [19] and Whitaker et al. [17], that interpreting the movement demands and generating suitable ascent-tactics was distinctly more arduous for the less experienced athletes from the NOV and INT groups than for the more accomplished athletes from the ADV group.

An interesting finding in Experiment 1 consists of the number of movement mistakes performed by the participants while ascending the task. Specifically, we could observe a negative correlation between the athletes' bouldering ability levels and their number of movement mistakes performed while attempting the task. Our findings emphasize that more accomplished participants performed a lower number of inappropriate or erroneous hand and feet actions that could have resulted in the failure to grasp a target handhold and would therefore have disabled them to complete the movement successfully. The lower number of movement mistakes entailed that more accomplished participants had to perform fewer ad hoc adaptations of their ascent-tactics since the target movements could more consistently be completed through their previously generated problem-solving tactics. In total, our findings emphasize, as previously discussed, that experienced participants could more accurately interpret the movement

demands and benefit from higher decision-making abilities, which contributed to more appropriate ascent-tactics among the ADV group. In contrast, video analysis revealed that less experienced participants frequently failed, for instance, to perceive the footholds E and F while attempting the task. The higher number of movement mistakes among the NOV and INT groups emphasizes that less experienced participants lacked to accurately inspect the movement demands, which contributed to more ad hoc adaptations of their initial ascent-tactics while attempting the task. Altogether, our findings highlight, in agreement with the previous findings by Sanchez et al. [16] and Pezzulo et al. [19], the pivotal role of movement repertoire and motor expertise to mentally simulate the movement sequences, to accurately interpret the movement demands, and to subsequently generate suitable ascent-tactics.

The lower number of movement deviations from the experts' best solution in the ADV group and the correlation between the athletes' bouldering ability levels and the number of movement deviations emphasize that the applied ascent-tactics of the ADV group followed the experts' best solution more accurately than those of the INT and NOV groups. However, we did not specifically examine the participants' choices for their ascent-tactics. Therefore, providing a rationale why the NOV and INT groups followed the experts' best solution less accurately remains intricate. Notwithstanding, we believe that physical constraints or motor action capabilities were less likely to solely explain the higher number of movements deviations among less experienced participants due to the design and the difficulty degree of the bouldering task. Furthermore, video analysis revealed that less experienced participants performed distinctly more trial-and-error movements, which underpins our previous assumption that they lacked to accurately interpret the movement demands during the bouldering preview. More precisely, common deviations from the best solution consisted of (a) grasping the handhold number 3 with the right hand, (b) keeping the left foot on the foothold C while attempting to grasp the handhold number 4, (c) ignoring the foothold E while grasping the handhold number 6, and (d) ignoring the foothold F while performing the last move to the top. In this context, it is worth mentioning that participants were informed, following the experiment, about the best solution of the task as perceived by experts and about their ascent-tactics accuracy while attempting the task. In consistency with the video analyses, participants from the NOV group predominantly reported that they focused, during the bouldering preview, mostly on the handholds and thus lacked accurately perceiving the footholds. This contributed, as previously discussed, to more inappropriate ascent-tactics and could explain the higher number of movement deviations from the experts' best solution among less-experienced participants.

The assumption of more suitable ascent-tactics among more experienced participants is furthermore corroborated by the lower average gripping and bouldering times to the top in the ADV group than in the INT and NOV groups. Although technical skills may have influenced the participants' motion fluidity, video analyses revealed that the lower bouldering times were mostly related to fewer non-movement times, which athletes generally require to interpret the movement demands while ascending the task. Higher motion fluidity in the ADV group, therefore, reflects more appropriate pre-ascent solving-tactics among more accomplished athletes. In contrast, participants from the NOV group took approximately twice as long as the ADV group to complete the task, which risked contributing, in agreement with Medernach et al. [4] and White and Olsen [13], to early muscular fatigue and thus to the failure to complete the task.

Our results revealed higher bouldering scores in the ADV group than in the NOV and INT groups, although the number of tops did not differ between the ADV and INT groups. Since high bouldering scores are in general attributable to multiple factors [1,32], it is intricate to exactly determine which factors could mostly explain our findings. A major explanation for

the better bouldering scores could be the significantly higher technical skills among more experienced participants. However, it is rather unlikely that technical skills could solely explain the higher bouldering scores due to the design and the difficulty degree of the task. For the same reason, higher grip strength among more experienced participants (i.e., non-significant differences between the INT and NOV groups) are also unlikely to explain our findings. This assumption is corroborated by the non-significant absolute grip strength differences we could observe between the NOV group (285.8 ± 51 N) and the elite female bouldering athletes of the retest procedure (290.5 ± 44 N; $p$ = .840) who could complete the task in their first attempts. In contrast, our findings concerning the movement deviations and movement mistakes emphasize that higher bouldering scores in the ADV group can be explained, at least to a large extent, by more appropriate ascent-tactics among more experienced participants. As previously discussed, more experienced participants could benefit from a more accurate interpretation of the movement demands during the bouldering preview and higher bouldering scores, therefore, reflect more suitable ascent-tactics and fewer movement mistakes among more accomplished participants.

## Conclusion

The findings in Experiment 1 revealed that more accomplished bouldering athletes were characterized by higher decision-making abilities and more suitable problem-solving tactics than less-experienced controls, contributing to fewer movement mistakes, shorter average gripping and bouldering times to the top, fewer movement deviations from the best solution, and higher bouldering scores. In total, the results in Experiment 1 emphasize that accomplished decision-making skills and suitable problem-solving tactics can be considered a key determinant in indoor bouldering. However, a major limitation in Experiment 1 consists of the relatively undemanding interpretation of the movement demands. That is why, in Experiment 2 we replicated the bouldering task of Experiment 1 by implementing slight handhold modifications. These handhold modifications increased the opportunity for misperceiving affordances and contributed, therefore, to a higher decision-making complexity.

## Experiment 2

In Experiment 2, we examined the participants' decision-making and problem-solving tactics while re-ascending the bouldering task of Experiment 1 with slight handhold modifications. These modifications made the perception of visual cues and potential affordances more challenging and contributed, therefore, to a higher decision-making complexity without, however, increasing the difficulty degree or distinctly modifying the movement demands compared to Experiment 1.

### Design of the bouldering task and assessment of the best solution

The three experts from Experiment 1 were tasked to implement the following handhold modifications: First, the handhold number seven was removed. Second, the handhold number four was replaced by a smaller handhold which was, however, set closer to the handhold number three. Third, experts added a total of three handholds (i.e., error grips), which were impossible to grasp and did not contribute to the completion of the task (Fig 3). In total, these handhold modifications did not substantially affect the movement demands (i.e., participants first had to grasp the top hold with their left hand) or the difficulty level of the task (i.e., 5B Fb) compared to Experiment 1. However, participants had to perceive during the bouldering preview that grasping the error grips did not contribute to the completion of the task and would increase the risk of falling off the wall. The generation of a suitable problem-solving tactic was thus

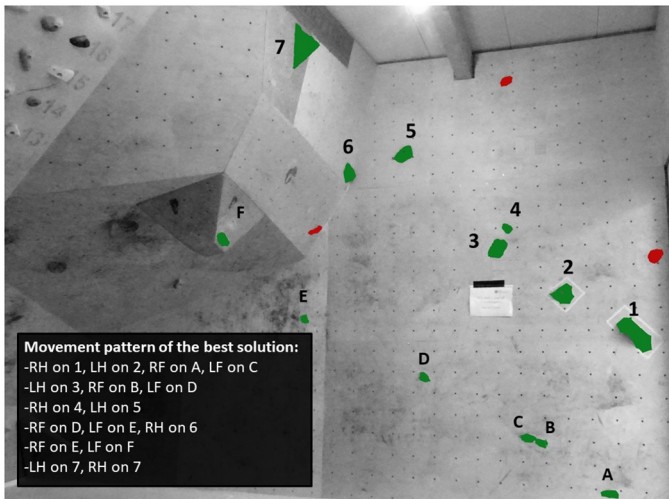

**Fig 3. The bouldering task in Experiment 2 (RH: Right hand; LH: Left hand; RF: Right foot; LF: Left foot).** In contrast to Experiment 1, the handhold number seven was removed and the handhold number four was replaced by a smaller handhold, which was set closer to the handhold number three. Experts also set a total of three error grips (i.e., marked in red colour), which were impossible to grasp and did not contribute to the completion of the task. The experts' best solution is displayed in the movement pattern.

more challenging since the error grips invited athletes to perform erroneous motor sequences. The procedure to determine the best solution of the task as perceived by the experts remained unchanged. The ICC indicated again perfect interrater reliability, with $r = 1$, and experts thus agreed on the best solution to complete the task. As in Experiment 1, the results of the retest procedure indicated non-significant differences for the best solution ($U = 7500$; $p = 0.111$; $r = 0.1$) between the experts and the female bouldering athletes who could, similar to Experiment 1, complete the bouldering task in their first attempts. These findings confirm, as in Experiment 1, the consistency of the ascent-tactic between the experts and the female bouldering athletes.

## Results

MANOVA revealed significant differences in the movement and decision-making abilities between the groups (see Table 3), with Pillai's trace $V = 1.171$, $F(16,130) = 11.48$, $p < 0.001$, $r = 0.8$; SP = 1.00. Specifically, the pre-ascent decision-making times, the number of movement mistakes and deviations from the best solution, and the average gripping and bouldering times to the top were significantly lower in the ADV group than in the NOV and INT groups. Relatedly, Spearman's rho indicated a significant negative correlation between the participants' bouldering ability levels and both the number of movement deviations from the best solution ($r = -0.809$; $p < 0.001$) and the number of movement mistakes ($r = -0.758$; $p < 0.001$). In contrast, the number of tops, the number of attempts to reach the top, and the number of error grips did not differ between the ADV group and the INT group. However, the number of tops was significantly lower and the number of error grips and attempts required to complete the task significantly higher in the NOV group than in both the INT and ADV groups. Participants from the NOV group demonstrated, in Experiment 2, shorter average gripping times ($Z = -2.796$, $p = 0.005$, $d = 0.8$), shorter bouldering times to the top ($Z = -2.582$, $p = 0.010$, $d = 0.9$), and fewer movement mistakes ($Z = -2.346$, $p = 0.019$, $d = 0.8$) than in Experiment 1. Furthermore, participants from the INT group performed, in Experiment 2, fewer movement

Table 3. Study results of the novice group (NOV), the intermediate group (INT), and the advanced group (ADV) in Experiment 2.

| | NOV (n = 18) | INT (n = 18) | ADV (n = 41) |
|---|---|---|---|
| decision-making times (s) | 99.7 ± 23 [88–111] $p < 0.001$; $d = 2.7$ | 41.3 ± 20 [31–51] $p = 0.012$; $d = 0.9$ | 27.4 ± 11 [24–31] $p < 0.001$; $d = 4.6$ |
| | | $F_{(2,76)} = 45.92$, $p < 0.001$, $r = 0.9$ | |
| deviations from the best solution (n) | 3.1 ± 0.7 [2.8–3.6] $p < 0.001$; $d = 1.8$ | 1.8 ± 0.7 [1.5–2.2] $p < 0.001$; $d = 1.4$ | 0.8 ± 0.7 [0.6–1.0] $p < 0.001$; $d = 3.2$ |
| | | $F_{(2,76)} = 64.58$, $p < 0.001$, $r = 0.8$ | |
| movement mistakes (n) | 3.0 ± 0.9 [2.5–3.4] $p < 0.001$; $d = 2.0$ | 1.2 ± 0.9 [0.7–1.6] $p = 0.004$; $d = 0.9$ | 0.4 ± 0.8 [0.1–0.6] $p < 0.001$; $d = 3.1$ |
| | | $F_{(2,76)} = 60.19$, $p < 0.001$, $r = 0.8$ | |
| average gripping times (s) | 4.7 ± 0.8 [4.3–5.1] $p = 0.140$ | 4.2 ± 0.9 [3.7–4.6] $p < 0.001$; $d = 1.5$ | 3.0 ± 0.7 [2.7–3.3] $p < 0.001$; $d = 2.3$ |
| | | $F_{(2,76)} = 37.27$, $p < 0.001$, $r = 0.7$ | |
| bouldering times to the top (s) | 37.8 ± 8 [34–42] $p < 0.001$; $d = 1.4$ | 27.2 ± 7 [23–33] $p < 0.001$; $d = 1.5$ | 18.8 ± 5 [17–21] $p < 0.001$; $d = 1.4$ |
| | | $F_{(2,70)} = 46.49$, $p < 0.001$, $r = 0.7$ | |
| number of tops (% / n) | 66.7 / 12; $p < 0.001$ | 100 / 18; $p = 1.00$ | 100 / 41; $p < 0.001$ |
| | | $F_{(2,70)} = 13.93$, $p < 0.001$, $r = 0.5$ | |
| number of attempts (n) | 3.6 ± 1.1 [3.0–4.1] $p < 0.001$; $d = 2.2$ | 1.4 ± 0.8 [1.0–1.8] $p = 0.152$ | 1.0 ± 0.0 [1–1] $p < 0.001$; $d = 4.3$ |
| | | $F_{(2,76)} = 87.58$, $p < 0.001$, $r = 0.8$ | |
| number of error grips (n) | 1.4 ± 1.1 [0.9–2.0] $p < 0.001$; $d = 0.9$ | 0.5 ± 0.8 [0.6–0.9] $p = 0.227$ | 0.1 ± 0.3 [0.0–0.2] $p < 0.001$; $d = 1.9$ |
| | | $F_{(2,76)} = 22.29$, $p < 0.001$, $r = 0.6$ | |

Results are given as mean (number or seconds) ± standard deviation with the 95% CI, except for the number of tops (percent and number). An alpha level of $p < 0.05$ was used to determine statistical significance.

mistakes ($Z = -2.355$, $p = 0.019$, $d = 0.9$) than in Experiment 1. Finally, participants from the ADV group demonstrated, in Experiment 2, lower decision-making times ($Z = -3.711$, $p < 0.001$, $d = 0.5$), fewer movement mistakes ($Z = -2.683$, $p = 0.007$, $d = 0.3$), and shorter average gripping ($Z = -2.054$, $p = 0.040$, $d = 0.1$) and bouldering times to the top ($Z = -4.342$, $p < 0.001$, $d = 0.5$) than in Experiment 1.

## Discussion

The purpose of Experiment 2 was to examine the participants' decision-making and problem-solving tactics while re-ascending the bouldering task of Experiment 1, though with slight handhold modifications that did not substantially affect the movement demands or the difficulty level of the task but made the generation of a suitable ascent-tactic more challenging compared to Experiment 1. The results in Experiment 2 are consistent with the previous findings in Experiment 1 and revealed that participants from the ADV group demonstrated shorter pre-ascent decision-making times, performed fewer movement mistakes and movement deviations from the best solution, and demonstrated shorter average gripping and bouldering times to the top than participants from the NOV and INT groups. In total, our findings emphasize, similar to Experiment 1, that participants from the ADV group could benefit from higher decision-making skills due to better movement knowledge, which contributed, in extension to Whitaker et al. [17], to a more accurate interpretation of the movement demands and more suitable problem-solving tactics to climb the task.

In contrast to Experiment 1, however, the INT and ADV groups demonstrated comparable bouldering scores with an equal number of tops and a comparable number of attempts to complete the task. These findings emphasize that participants from the ADV group were not characterized by better bouldering scores despite more suitable ascent-tactics with fewer movement deviations and movement mistakes. Yet, it must be considered that the difficulty degree of the task was distinctly below the bouldering ability levels of the ADV group and the INT group. Therefore, participants from both groups could mostly attempt the task in their first attempts, which explains the non-significant findings between the INT and ADV groups. Relatedly, both groups were furthermore characterized by a similar number of error grips applied by participants while attempting the task, highlighting that both groups could successfully perceive, during the bouldering preview, that grasping the error grips did not contribute to a successful completion of the task. In contrast, participants from the NOV group integrated a significantly higher number of error grips into their ascent-tactics. Specifically, they grasped, on average, one of the three error grips while attempting the bouldering task, whereas participants from the ADV group predominantly ignored the error grips. These findings underpin our assumptions that less experienced participants lacked to accurately interpret the movement demands and that more successful athletes could benefit from higher decision-making abilities and thus generate more suitable ascent-tactics.

An interesting finding in Experiment 2 consists of the fewer movement mistakes performed by the three ability groups compared to Experiment 1. Relatedly, participants from the NOV and ADV groups furthermore demonstrated in Experiment 2 shorter average gripping and bouldering times to the top, and participants from the ADV group required in Experiment 2 shorter decision-making times than in Experiment 1. Although we did not observe improvements in all independent variables, our findings emphasize that participants could rely on their problem-solving tactics in Experiment 1 to generate their ascent-tactics in the present Experiment 2. Participants were, therefore, able to benefit from a broader movement repertoire acquired in Experiment 1 to make anticipatory decisions and to generate appropriate ascent-tactics in Experiment 2. Our findings highlight, in agreement with Sanchez et al. [16], the pivotal role of movement repertoire from earlier sport-specific experiences in accomplished decision-making performances. Additionally, the findings are consistent with the previous observations by Ferrand, Tetard, and Fontayne [33] who described a lack of climbing route knowledge as one type of self-handicapping in sport climbing competitions.

## Conclusion

The findings in Experiment 2 are consistent with the results in Experiment 1 and revealed that more accomplished bouldering athletes were characterized by higher decision-making abilities and more suitable problem-solving tactics than less-experienced controls, contributing to fewer movement mistakes, shorter average gripping and bouldering times to the top, and fewer movement deviations from the best solution. In total, the results in Experiment 2 emphasize, in consistency with Experiment 1, that accomplished decision-making skills and suitable problem-solving tactics consist of a key determinant in indoor bouldering. However, although the implementation of slight handhold modifications made the perception of visual cues and potential affordances more challenging, interpreting the movement demands was still relatively undemanding and we, therefore, implemented in Experiment 3 a bouldering task with multiple ascent-solutions that contributed to a distinctly higher decision-making complexity.

## Experiment 3

In Experiment 3, participants were exposed to a novel bouldering task with many climbing holds varying in size, shape, and colour. Although the climbing hold arrangement permitted multiple ascent-solutions, most handholds were hard to grasp and participants had thus to accurately select, during the bouldering preview, the well-shaped handholds to generate appropriate ascent-tactics.

### Design of the bouldering task and assessment of the best solution

In the present experiment, we were curious about whether participants would be able to perceive and thus to integrate into their ascent-tactics the well-shaped handholds that were undemanding to grasp. Therefore, and in contrast to the previous experiments, experts set many climbing holds varying in size, shape, and colour on a 3.0-m high and 25-degree overhanging bouldering wall (Fig 4). Furthermore, they only marked the two handholds of the starting position (i.e., handhold number 1 and 2) and the last hold, which marked the top view of the task (i.e., handhold number 9). Between these marked handholds, participants were free to integrate any hand- and footholds into their ascent-tactics. This contributed to a distinct higher decision-making complexity compared to the previous experiments as participants had to accurately select the well-shaped handholds and reject those handholds that were hard to grasp. To avoid that footholds would impede participants from integrating the well-shaped handholds into their ascent-tactics, the setup offered well-sized footholds that were undemanding to step on. Consequently, grasping a particular handhold (e.g., the handhold number 4) could occur using distinct footholds relative to the athletes' physical constraints and motor action capabilities. Since footholds would thus not affect the participants' choices whether to integrate or reject a specific handhold, they were excluded from the best solution and from the number of movement deviations participants performed while attempting the task. This means that we did not evaluate the footholds participants integrated into their ascent-tactics and the best solution was restricted to the sequence of handholds that experts perceived as the most straightforward ascent-path. Inappropriate or erroneous feet actions were still, similar to the previous experiments, examined through the number of movement mistakes.

The handhold arrangement in the experts' best solution was suggested to have a difficulty degree of approximately 5B Fb and thus comparable to the previous experiments. The ICC indicated, similar to the previous experiments, a high correlation with $r = 0.949$ and could consequently confirm the consistency between experts. As in the previous experiments, results of the retest procedure indicated non-significant differences for the best solution ($U = 7500$; $p = 0.055$; $r = 0.2$) between the experts and the female bouldering athletes. However, given that the $p$-value was very close to 0.05, we decided to more profoundly analyse the ascent-tactics of the female bouldering athletes. Specifically, video analyses revealed that the female athletes mostly followed the experts' best solution with, however, some athletes using in the middle section of the task different handholds than those in the experts' best solution. More precisely, we found that female athletes who did not solely followed the experts' best solution integrated the same handholds into their alternative ascent-tactics. These handholds, hereinafter referred to as 'key grips', were, similar to those in the best solution, well-shaped and unchallenging to grasp and were, therefore, considered to be a valid alternative to the route setters' best solution.

### Results

MANOVA revealed significant differences in the movement and decision-making abilities between the groups (see Table 4), with Pillai's trace $V = 1.234$, $F(18,130) = 11.64$, $p < 0.001$,

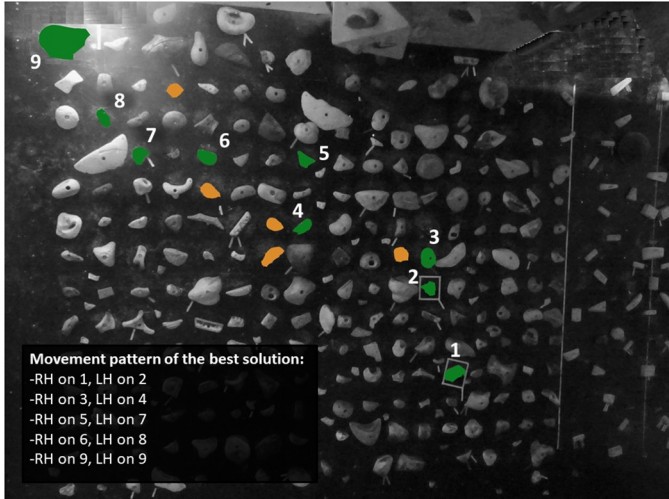

**Fig 4. The bouldering task in Experiment 3 (RH: Right hand; LH: Left hand).** The task was set with many climbing holds varying in size, shape, and colour. Experts marked the two handholds of the starting position (RH on handhold number 1 and LH on handhold number 2) and the last hold (handhold number 9). Between these marked handholds, participants were free to integrate any handholds into their ascent-tactics. The handholds of the experts' best solution are marked in green colour and key grips (i.e., not part of the best solution but unchallenging to grasp and thus considered as valid alternative ascent-tactic) in orange colour.

$r = 0.8$, SP = 1.00. Specifically, the pre-ascent decision-making times, the number of movement mistakes, and the average gripping and bouldering times to the top were significantly lower in the ADV group than in the NOV and INT groups. In contrast, the number of movement deviations from the best solution, the number of tops, and the number of attempts to reach the top did not differ between the ADV group and the INT group. However, the number of tops were significantly lower and the number of movement deviations and attempts required to complete the task significantly higher in the NOV group than in the INT group and the ADV group. Relatedly, Spearman's rho indicated a significant negative correlation between the bouldering ability levels of the participants and both the number of movement deviations from the best solution ($r = -0.675$; $p < 0.001$) and the number of movement mistakes ($r = -0.785$; $p < 0.001$). Furthermore, the total number of handholds used by the participants to complete the task was significantly lower in the ADV group than in the INT and NOV groups. In contrast, however, the number of key grips used by the participants did not differ between the groups.

## Discussion

In Experiment 3, we examined the participants' decision-making skills by implementing a bouldering task with many climbing holds varying in size, shape, and colour. Although the climbing hold arrangement permitted multiple ascent-solutions, most handholds were hard to grasp and we were curious about whether participants would perceive, during the bouldering preview, and subsequently integrate into their ascent-tactics, the well-shaped handholds that were undemanding to grasp and reject those handholds that were hard to grasp.

Our findings in Experiment 3 are consistent with the results in the previous experiments and revealed that participants from the ADV group demonstrated shorter pre-ascent decision-making times, performed fewer movement mistakes, and demonstrated shorter average gripping and bouldering times to the top than participants from the NOV and INT groups. In

**Table 4. Study results of the novice group (NOV), the intermediate group (INT), and the advanced group (ADV) in Experiment 3.**

| | NOV ($n = 18$) | INT ($n = 18$) | ADV ($n = 41$) |
|---|---|---|---|
| decision-making times ($s$) | 98.9 ± 32 [83–115] $p < 0.001$; $d = 1.2$ | 63.4 ± 25 [51–76] $p = 0.003$; $d = 1.1$ | 39.7 ± 19 [34–45] $p < 0.001$; $d = 1.5$ |
| | $F(2,76) = 37.82$, $p < 0.001$, $r = 0.7$ | | |
| deviations from the best solution ($n$) | 10.4 ± 2.2 [9.3–11.0] $p < 0.001$; $d = 2.6$ | 4.4 ± 2.4 [3.2–5.6] $p = 0.263$ | 3.3 ± 2.2 [2.6–4.0] $p < 0.001$; $d = 3.2$ |
| | $F(2,76) = 62.65$, $p < 0.001$, $r = 0.8$ | | |
| movement mistakes ($n$) | 5.8 ± 1.7 [5.0–6.7] $p < 0.001$; $d = 2.1$ | 2.4 ± 1.5 [1.6–3.2] $p = 0.001$; $d = 1.1$ | 0.9 ± 1.3 [0.1–1.3] $p < 0.001$; $d = 3.4$ |
| | $F(2,76) = 74.74$, $p < 0.001$, $r = 0.8$ | | |
| average gripping times ($s$) | 5.0 ± 1.2 [4.4–5.6] $p = 0.069$ | 4.3 ± 1.1 [3.7–4.8] $p = 0.003$; $d = 1.1$ | 3.3 ± 0.8 [3.1–3.6] $p < 0.001$; $d = 1.8$ |
| | $F(2,76) = 20.24$, $p < 0.001$, $r = 0.6$ | | |
| bouldering times to the top ($s$) | 49.3 ± 9 [45–54] $p = 0.001$; $d = 1.1$ | 37.6 ± 12 [31–44] $p < 0.001$; $d = 1.6$ | 22.8 ± 8 [20–25] $p < 0.001$; $d = 3.2$ |
| | $F(2,68) = 52.40$, $p < 0.001$, $r = .8$ | | |
| number of tops ($\% / n$) | 55.6 / 10 $p < 0.001$ | 94.4 / 17 $p = 0.835$ | 100 / 41 $p < 0.001$ |
| | $F(2,68) = 43.05$, $p < 0.001$, $r = 0.7$ | | |
| number of attempts ($n$) | 4.9 ± 1.5 [4.1–5.6] $p < 0.001$; $d = 2.9$ | 1.4 ± 0.7 [1.1–1.8] $p = 0.371$ | 1.1 ± 0.3 [1.0–1.2] $p < 0.001$; $d = 4.3$ |
| | $F(2,76) = 135.75$, $p < 0.001$, $r = 0.8$ | | |
| number of key grips ($n$) | 3.0 ± 1.5 [2.3–3.7] $p = 0.980$ | 2.9 ± 0.8 [2.5–3.4] $p = 0.158$ | 2.3 ± 1.0 [1.9–2.6] $p = 0.106$ |
| | $F(2,76) = 3.26$, $p = 0.055$ | | |
| total number of handholds ($n$) | 15.1 ± 3 [14–16] $p < 0.001$; $d = 1.3$ | 11.1 ± 3 [9–13] $p = 0.003$; $d = 1.1$ | 8.9 ± 1.2 [8.5–9.3] $p < 0.001$; $d = 3.2$ |
| | $F(2,68) = 55.04$, $p < 0.001$, $r = 0.8$ | | |

Results are given as mean (number or seconds) ± standard deviation with the 95% CI, except for the number of tops (percent and number). An alpha level of $p < 0.05$ was used to determine statistical significance.

total, our findings emphasize, similar to the previous experiments, that participants from the ADV group could benefit from higher decision-making skills due to better movement knowledge, which contributed to a more accurate interpretation of the movement demands and more suitable problem-solving tactics to climb the task.

Similar to the previous experiments, participants from the ADV group performed, regardless how accurately they followed the route setters' best solution, fewer movement mistakes. The ADV group thus made fewer fails to grasp a target handhold and managed to complete a particular movement more successfully, in contrast to participants from the NOV and INT groups. This means that the more accomplished participants from the ADV group demonstrated fewer ad hoc adaptations of their ascent-tactics as they could, more consistently, complete the task through their previously generated problem-solving tactics. Our findings, therefore, emphasize more suitable ascent-tactics among the ADV group. This assumption is furthermore underpinned by the shorter average gripping and bouldering times to the top in the ADV group than in the less experienced control groups. Specifically, as a result of more suitable ascent-tactics and fewer movement mistakes, participants from the ADV group demonstrated higher motion fluidity and shorter non-movement times, which athletes generally require to interpret the movement demands or adapt their ascent-tactics while ascending the

task. In contrast, and similar to the previous experiments, video analyses revealed that participants from the NOV and INT groups predominantly applied trial-and-error approaches to complete the task, which contributed to a higher number of movement mistakes and longer average gripping and bouldering times to the top.

As in the previous experiments, we found a negative correlation between the athletes' bouldering ability levels and their number of movement deviations from the route setters' best solution. Specifically, the number of movement deviations from the best solution was higher in the NOV group than in the INT group and the ADV group. These results emphasize that the ADV and INT groups followed the experts' sequence of handholds more accurately than participants from the NOV group. However, and in contrast to the previous experiments, the number of movement deviations from the best solution did not differ between the ADV group and the INT group. These findings suggest that participants from the ADV group did not outperform participants from the INT group in perceiving the well-shaped handholds experts integrated into their best solution. An interesting finding in Experiment 3, however, was that all three ability groups demonstrated a significantly higher number of movement deviations from the best solution than in the previous experiments. Although these findings were to be expected as the climbing hold arrangement permitted multiple ascent-solutions and invited athletes to seek versatile ascent-tactics, video analyses furthermore revealed that all groups integrated a relative high number of key grips into their ascent-tactics with non-significant differences between the ability groups. Since participants from the ADV group were characterized by high bouldering scores, a low number of movement mistakes, and a low number of total handholds to ascent the task, it is less likely that they lacked perceiving the well-shaped handholds of the best solution. In fact, our results emphasize that they deliberately integrated the key grips into their ascent-tactics and consciously rejected some of the handholds experts integrated into their best solution. This contributed to a higher number of movement deviations from the best solution and could explain the non-significant results between the INT group and the ADV group. It is conceivable, consistent with the ascent-tactics of the female athletes in the retest procedure, that using the key grips did not just consist of a valid alternative to the experts' best solution. Indeed, the use of the key grips could even have contributed to a more straightforward ascent-tactic for non-elite bouldering athletes, instead of solely following the best solution as perceived by the experts. In this context, it is worth mentioning that the bouldering task was set on a 25-degree overhanging bouldering wall and that less experienced participants may, therefore, have been impeded, in agreement with Whitaker et al. [17] and Sanchez et al. [26], by their motor skills (e.g., upper body power or lock-off ability, which are both physical determinants, although not assessed in the present investigation) to follow the best solution of the experts. In total, it is questionable whether the experts' best solution consisted of the most straightforward ascent-path and thus bestowed the solution for all. This is especially relevant for the less experienced athletes. Consequently, the number of movement deviations from the best solution does not provide a rationale of better decision-making and more appropriate ascent-tactics among more experienced participants.

Relatedly, participants from the ADV and INT group were characterized by comparable bouldering scores with non-significant differences in the number of tops and the number of attempts required to complete the task. These findings suggest that more suitable ascent-tactics did not contribute to higher bouldering scores as participants from the ADV group did not outperform participants from the INT group. However, as in Experiment 2, the difficulty degree of the bouldering task was again distinctly below the bouldering ability levels of both groups and participants could thus mostly complete the task in their first attempts.

## Conclusion

The results in Experiment 3 revealed that more accomplished bouldering athletes were characterized by higher decision-making abilities and more suitable problem-solving tactics than less-experienced controls, contributing to fewer movement mistakes, shorter average gripping and bouldering times to the top, and a lower number of handholds required to complete the task. In total, our findings emphasize, in consistency with the previous experiments, that accomplished decision-making skills and suitable problem-solving tactics consist of a key determinant in indoor bouldering. Our results furthermore suggest that bouldering tasks with multiple ascent-solutions and higher decision-making complexity invite athletes in their decision-making processes to seek alternative ascent-tactics relative to their physical constraints and motor action capabilities. Generating suitable ascent-tactics that best fit the task, can substantially vary on personal characteristics. The concept of a general best solution is thus questionable, particularly in bouldering tasks with higher movement complexity as in Experiment 3.

## General discussion

The purpose of the present research program was to investigate the effects of decision-making on indoor bouldering performances by implementing a multi-experimental and ecologically valid study design with the conceptual replication of three bouldering tasks. Specifically, our study aimed to examine whether novice, intermediate, and advanced bouldering athletes would differ in their decision-making abilities and to what extend distinct problem-solving tactics would affect the athletes' bouldering performances. To the best of our knowledge, this is the first study to investigate decision-making performances and problem-solving tactics in indoor bouldering athletes.

In summary, the results across the three experiments highlight, in agreement with Sanchez et al. [16], that accomplished decision-making abilities and efficient problem-solving tactics can be considered, among other factors such as technical skills and conditioning aspects [1,32], a key determinant of success in indoor bouldering. We could observe across the three experiments that novice, intermediate, and advanced bouldering athletes applied distinct problem-solving tactics with more appropriate ascent-tactics among more experienced athletes. Specifically, our results revealed that advanced bouldering athletes required shorter pre-ascent decision-making times (Experiment 1, 2, & 3), performed a lower number of movement mistakes (Experiment 1, 2, & 3) and movement deviations from the best solution (Experiment 1 & 2), and demonstrated shorter average gripping and bouldering times to the top (Experiment 1, 2, & 3) than novice and intermediate athletes. Relatedly, intermediate bouldering athletes required shorter pre-ascent decision-making times (Experiment 1, 2, & 3), performed a lower number of movement mistakes (Experiment 1, 2, & 3) and movement deviations from the best solution (Experiment 1, 2, & 3), demonstrated shorter average gripping (Experiment 1) and bouldering times to the top (Experiment 1, 2, & 3), and were characterized by higher bouldering scores (Experiment 1, 2, & 3) compared to novice athletes.

In agreement with Sanchez et al. [16] and Whitaker et al. [17], better decision-making performances and more suitable problem-solving tactics were related to a more appropriate bouldering preview among experienced participants. Specifically, our findings suggest that experienced bouldering athletes could benefit, during the bouldering preview, from a better perception of visual cues and a more accurate identification of affordances. This contributed to a more appropriate interpretation of the movement demands and a better rejection of ineffective movement solutions. Our results are consistent with the previous findings of Pezzulo et al. [19] who observed that experienced climbers demonstrated a better recall of climbing hold positions following a route preview as a result of higher motor skills, and Whitaker et al.

[17] who found that climbing expertise was positively associated with the perceptual judgement accuracy of affordances and visual memory abilities. Our results are furthermore consistent with recent findings from team sports, which suggest that the perception of advance cues positively affects the players' anticipatory behaviour and decision-making performances [34]. Roca, Ford, and Memmert [35], for instance, examined the decision-making performances of soccer players during simulated attacking situations and observed that experienced players employed a broader attentional focus and spent more time fixating on attackers in threatening positions than less-experienced players, highlighting the pivotal role of visual search strategies for appropriate decision-making performances in soccer.

The findings of the present research program furthermore emphasize that a high movement repertoire, which permits experienced athletes to accurately compare the movement demands of the task with previously experienced stimuli, is propitious to accurately interpret the perceptual information from the bouldering preview [16]. As a result of a higher movement repertoire, more experienced bouldering athletes benefit from better anticipatory decisions and more appropriate problem-solving tactics relative to the task [16,20]. Although previously not mentioned, we observed within the ADV group in all three experiments that participants with previous bouldering competition experiences (i.e., over at least one year) were characterized by distinctly shorter decision-making times, fewer movement deviations from the best solution, and shorter bouldering times to the top than participants with comparable ability levels but no previous competition experiences. These findings furthermore emphasize the pivotal role of movement repertoire as a result of long-term engagement in deliberate practice and competitive bouldering [8,16,18].

Although the present research program could highlight that effective decision-making performances and appropriate problem-solving tactics can be considered a key determinant of success in indoor bouldering, several limitations must be considered. First, we gave preference, particularly in Experiment 1 and Experiment 2, to bouldering tasks with relatively low difficulty degrees and undemanding movements to ensure that the tasks were technically and physically climbable by all participants and to minimize the impact of interfering factors such as physical constraints (e.g., body height) or motor capabilities (e.g., grip strength) on the participants' decision-making performances. Consequently, our experimental design contributed to high bouldering scores in the INT and ADV groups among the three experiments. More precisely, the INT and ADV groups demonstrated, in all three experiments, comparable bouldering scores with a similar number of tops and a comparable number of attempts to complete the tasks. These findings suggest that more suitable ascent-tactics did not contribute to higher bouldering scores as participants from the ADV group did not outperform participants from the INT group. Therefore, further empirical investigations are needed in order to examine the athletes' decision-making and problem-solving tactics in bouldering tasks with distinctly higher difficulty degrees and substantially higher movement diversity and complexity (e.g., athletic movements, coordinative movements). This would provide additional knowledge and would contribute to a more holistic understanding concerning decision-making in indoor bouldering.

Future research must also more profoundly examine the decision-making performances of bouldering athletes with comparable ability levels to provide additional knowledge concerning their problem-solving tactics. As previously mentioned, we observed better decision-making abilities in participants with previous competition experiences and it could, therefore, be interesting to examine the ascent-tactics of world-class athletes during a bouldering world cup. In this context, it could be worthwhile to examine whether athletes would apply the same ascent-tactics or rely on alternative ascent solutions, particularly when they fail to complete a bouldering task in their first attempts. Relatedly, female participants were excluded in the present

research program from data and future research could examine whether male and female bouldering athletes would differ in their ascent-tactics while attempting the same tasks.

As previously discussed, our results in Experiment 3 revealed that the number of movement deviations from the best solution did not differ between the ADV and the INT group. The differential findings across the experiments concerning the movement deviations (i.e., in Experiment 1 and 2, athletes from the ADV group outperformed both the INT and NOV groups, while in Experiment 3 non-significant results were found between the ADV and INT groups) can mostly be related to the movement demands of the bouldering tasks. Specifically, the setting of the tasks in Experiment 1 and Experiment 2 (i.e., low difficulty degree, positively shaped climbing holds, low distances between the handholds) contributed to one apparent climbing path, which was best completed by following the experts' best solution. However, the findings in Experiment 3 emphasize that pursuing solely the experts' best solution did not necessarily contribute to the most straightforward ascent-path, particularly for lower skilled bouldering athletes. Therefore, the concept of a general best solution that fits all athletes best is questionable since the best solution as perceived by experts does not necessarily consist of the most straightforward ascent-path for less experienced athletes, particularly in bouldering tasks with higher movement complexity that invite athletes to seek alternative ascent-tactics relative to their physical constraints and motor action capabilities. Relatedly, and as mentioned in the discussion section of Experiment 1, we did not specifically examine the participants' choices of their ascent-tactics and could thus not provide a rationale why athletes applied the same ascent-tactics as the experts or relied on alternative ascent-solutions. Although participants provided information on their decision-making choices following the three experiments, thinking aloud and more structured dialogues following the decision-making period could have provided additional information about the athletes' decision-making. Such additional information about the athletes' choices could also have contributed to a better link between internal cognitive processes of decision-making and the external measurements of the bouldering performances.

Finally, a further limitation of the present study consists in the statistical analyses. First, the unbalanced distribution of participants between the groups must be considered. More precisely, athletes were randomly recruited to provide a high number of participants, which contributed to a distinctly lower number of participants in the novice and intermediate groups and could have biased the statistical outcomes. Furthermore, using the same sample across the three experiments contributed to essential insights regarding the athletes' decision-making performances under various experimental conditions. However, the use of the same sample across multiple comparisons may also have increased the risk of type I errors due to the relatively high number of statistical tests and a non-adjusted alpha level.

## Practical applications

Although further research on cognitive processes in indoor bouldering is requested, the results in the present study suggest, in accordance with Sanchez et al. [16] and Whitaker et al. [17], that bouldering regimen should implement specific exercises that foster the athletes' bouldering preview skills. Our results furthermore suggest, in agreement with Sanchez et al. [16] and Ferrand et al. [33], that athletes should seek a high movement repertoire through (a) deliberate practice, (b) repeating bouldering tasks with comparable movement demands, and (c) bouldering competitions. In addition, studies in the field of team sports emphasize that the involvement in a diverse range of sports and physical activities could be valuable for the development of the athletes' decision-making skills [36]. As these studies emphasize that cognitive skill acquisitions are transferable across sport activities with similar structures [36], we suggest that

the deliberate practice of rock and sport climbing could be beneficial to foster the athletes' the decision-making abilities in indoor bouldering.

## Overall conclusion

Despite several limitations, the present research program could highlight that accomplished decision-making abilities consist, among other factors, of a key determinant in successful indoor bouldering performances. Particularly bouldering competitors must, due to relative high time constraints, strive to promptly identify suitable problem-solving tactics and constantly adapt their ascent-tactics to the specific movement demands of the tasks. We could provide evidence by implementing a multi-experimental study design with the conceptual replication of three bouldering tasks that more experienced and higher-skilled bouldering athletes required shorter decision-making times to generate more suitable problem-solving tactics as a result of a more appropriate bouldering preview and higher movement repertoire. However, further empirical investigations are compelling to provide additional knowledge and to contribute to a more holistic understanding concerning the pivotal role of decision-making in indoor bouldering.

## Supporting information

**S1 Fig. Full MANOVA reports of the Experiment 1.**
(TIF)

**S2 Fig. Full MANOVA reports of the Experiment 2.**
(TIF)

**S3 Fig. Full MANOVA reports of the Experiment 3.**
(TIF)

## Acknowledgments

We would like to thank all participants who volunteered for this study.

## Author Contributions

**Conceptualization:** Jerry Prosper Medernach, Daniel Memmert.

**Data curation:** Jerry Prosper Medernach, Daniel Memmert.

**Investigation:** Jerry Prosper Medernach.

**Methodology:** Jerry Prosper Medernach, Daniel Memmert.

**Supervision:** Daniel Memmert.

**Validation:** Daniel Memmert.

**Writing – original draft:** Jerry Prosper Medernach.

**Writing – review & editing:** Daniel Memmert.

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
