## [Decision Letter · Decision Letter 0]

16 Dec 2020

PONE-D-20-24969

Effects of Decision-making on Indoor Bouldering Performances: A multi-experimental Study Approach

PLOS ONE

Dear Dr. Medernach,

Thank you for submitting your manuscript to PLOS ONE. After careful consideration, we feel that it has merit but does not fully meet PLOS ONE’s publication criteria as it currently stands. Therefore, we invite you to submit a revised version of the manuscript that addresses the points raised during the review process.

Please, address point-to-point all reviewers' issues (in particular Reviewer 1's ones).

We look forward to receiving your revised manuscript.

Kind regards,

Luca Paolo Ardigò, Ph.D.

Academic Editor

PLOS ONE

Journal Requirements:

Additional Editor Comments:

Please, address point-to-point all reviewers' issues (in particular Reviewer 1's ones).

Reviewers' comments:

Reviewer's Responses to Questions

**Comments to the Author**

1. Is the manuscript technically sound, and do the data support the conclusions?

Reviewer #1: Partly

Reviewer #2: Yes

Reviewer #3: Yes

2. Has the statistical analysis been performed appropriately and rigorously? 

Reviewer #1: No

Reviewer #2: I Don't Know

Reviewer #3: Yes

3. Have the authors made all data underlying the findings in their manuscript fully available?

Reviewer #1: Yes

Reviewer #2: No

Reviewer #3: No

4. Is the manuscript presented in an intelligible fashion and written in standard English?

Reviewer #1: Yes

Reviewer #2: Yes

Reviewer #3: Yes

5. Review Comments to the Author

Reviewer #1: I think this is an interesting and well-motivated study. Based on my understanding across three experiments, intermediates and advanced climbers, compared to novices, had a higher number of successful attempts (Experiment 1, 2, & 3), used less observation time (Experiment 1, 2, & 3), had fewer total attempts (Experiment 2 & 3), used fewer error grips (Experiment 2), and had fewer movement deviations (Experiment 3); advanced climbers, compared to novices and intermediates, had fewer total attempts (Experiment 1), had fewer movement deviations (Experiment 1 & 2), took less time to ascend the route (Experiment 1, 2, & 3), gripped holds for less time (Experiment 1, 2, & 3), and made fewer movement mistakes (Experiment 1 & 2; in Experiment 3 advanced climbers made fewer movement mistakes than intermediate climbers, and intermediate climbers made fewer movement mistakes than novice climbers). No differences were found between groups for the number of key grips used (Experiment 3). The authors provide well-grounded descriptions of decision-making in sport, and how it applies to bouldering. Further, I appreciated the authors thoughtful consideration of the limitations of their study. However, I had some concerns about the statistical approach that I think should be addressed, as well as some smaller points about literature and other miscellaneous issues. I would recommend a rejection of the manuscript as is with an invitation to re-submit if the issues below can be addressed.

Major Concerns:

I was concerned with the number of statistical tests employed in this experiment, and the fact that the same sample was used across three experiments. I think at minimum implications for Type I errors should be discussed in the limitations section. The authors might also consider adjusting their alpha level to account for multiple comparisons.

Additionally, I was confused as to why ANOVA was chosen over linear regression to analyze the data. There are not natural divisions or groupings across skill level in climbing (and therefore the group divisions are always somewhat arbitrary), and regression would prevent arbitrary divisions in the data set. The findings in this study were not always consistent across skill level (i.e. sometimes experts outperformed both intermediates and novices, while other times experts and intermediates outperformed novices), this made the results hard to track and confusing to interpret. I think that retaining the continuous nature of the data with regression might be more informative. Further, it is harder to intuitively interpret the size of effects in ANOVA and I think using linear regression would allow for a more intuitive interpretation of the data. If ANOVA is to be used, this statistical choice should be justified, and the differential findings across groups should be further discussed.

While I think the authors do a good job discussing decision making in sport, I think they should spend more time connecting how their measures assess decision making, as I felt this wasn’t completely clear.

Minor Comments:

I found tables in the results section useful, but overall the results section was rather cumbersome to read, and I felt interpretation of effect sizes would help break up the number of statistical tests presented, as well as help the reader understand the practical significance of the findings.

p. 8 line 199, affordances should be defined and Gibson should be cited. Further, I think the authors should spend more time developing how affordances relate to the theoretical concepts they are discussing and the study they are presenting.

Relatedly, I would explain how affordances were made more challenging. My understanding of Gibson’s view of affordances is that we either perceive an opportunity for action or we don’t. Affordances are the interaction between the environment and the observer, they do not reside in the environment alone.

There were some sentences throughout that were awkwardly worded (p. 4 lines 99-108, p.6 lines 143-145 & lines 159-162).

I might spend a bit of time justifying why bouldering is an open-skill sport. I would think it is probably somewhere on the continuum from open to closed, as many aspects of the environment are fixed in bouldering. The authors may just want to add a bit more nuance to the discussion of open/closed skill in the context of bouldering.

I would make sure to include effect sizes for each statistical test. For example, on p. 17 lines 438-442. These are potentially interesting results, but an effect size would aid in interpretation of these findings.

On page 15 line 193, why isn’t an r value included for this comparison?

Reviewer #2: The authors examined decision-making performance in bouldering. Across three different tasks, the authors found that level of expertise (novice, intermediate, advanced) related to different decision-making and performance outcomes.

I found the paper very interesting and thought that it makes a useful contribution to the literature.

I had several minor suggestions that I hope might improve the paper:

1. The authors could note more clearly the covariates used in the statistical analysis section (e.g., did authors control for grip strength?) and report full ANOVA/MANOVA tables as an appendix

2. The authors could consider removing statistics from their abstract, which might shorten it and make it easier to read

3. The authors could give a very brief overview of three experiments, how they differ, and the their goals in the final paragraph of the introduction

4. The authors could very briefly explain the grading system for those not familiar with it

5. The authors could explain the p-values in the tables both in the text and in the legend to each table

6. The authors could confirm data availability as per this journal’s policy

7. The authors could clarify the role of physical strength – apart from grip strength – in the sport (e.g., pull up strength) and whether this could play a role at a novice/intermediate level (if so, the authors could note if this was assessed)

8. The authors could expand their discussion on the significance of findings for general cognitive theories of expertise beyond bouldering.

Reviewer #3: Introduction

Generally, well-written and scientifically supported by a good rationale.

Line 173: It would be interesting to read a clear statement of contribution before the study purpose.

Methods

Line 211: Explain how the study size was arrived at. Possibly, information about sample size calculation can be moved from statistics to this section. Just a suggestion.

Lines 211-213: Describe the setting, locations, and relevant dates, including periods of recruitment, exposure, follow-up, and data collection

Lines 213-215: provide the sources and methods of selection of participants

Line 287: provide a structure in the section would help to read. Possibly, organizing by “anthropometrics and strength” and “Movement analysis” or similar

Lines 313-326: who made the analysis? Specify the characteristics of the observer. Additionally, how accuracy and precision of data collected was ensured? Please provide information about validity and reliability of the approach.

Results

Effect size is missing in the tables. Any possibility to include?

Text in results is sometimes repeating the information in the tables. Not necessary and can be considered to simplify.

Discussion

At the end of each discussion, a section of summary of findings, particular practical applications (for each study), and future research directions should be added, as well as identification of study limitations.

6. PLOS authors have the option to publish the peer review history of their article (what does this mean?). If published, this will include your full peer review and any attached files.

Reviewer #1: No

Reviewer #2: **Yes: **Michael Connors

Reviewer #3: No

---

## [Author Response · Author response to Decision Letter 0]

28 Jan 2021

Dear Academic Editor and reviewers, 

We would like to thank you for the useful comments and suggestions you made on our manuscript. Your comments were undoubtedly helpful to improve the quality of our paper. 

Please refer to our response to the reviewers letter in the Attach Files Section. 

As you hopefully will notice by reading our response, we strove to implement all the recommendations made by the reviewers and to address point-to-point the reviewers’ issues. 

We, therefore, hope that our revised manuscript now fully meets PLOS ONE´s publication criteria.

We are looking forward to your response.

Kind regards,

Jerry Medernach

---

## [Decision Letter · Decision Letter 1]

25 Feb 2021

PONE-D-20-24969R1

Effects of Decision-making on Indoor Bouldering Performances: A multi-experimental Study Approach

PLOS ONE

Dear Dr. Medernach,

Thank you for submitting your manuscript to PLOS ONE. After careful consideration, we feel that it has merit but does not fully meet PLOS ONE’s publication criteria as it currently stands. Therefore, we invite you to submit a revised version of the manuscript that addresses the points raised during the review process.

Please, one further effort to address Reviewer 1's minor issues.

We look forward to receiving your revised manuscript.

Kind regards,

Luca Paolo Ardigò, Ph.D.

Academic Editor

PLOS ONE

Journal Requirements:

Additional Editor Comments (if provided):

Please, one further effort to address Reviewer 1's minor issues.

Reviewers' comments:

Reviewer's Responses to Questions

**Comments to the Author**

1. If the authors have adequately addressed your comments raised in a previous round of review and you feel that this manuscript is now acceptable for publication, you may indicate that here to bypass the “Comments to the Author” section, enter your conflict of interest statement in the “Confidential to Editor” section, and submit your "Accept" recommendation.

Reviewer #1: (No Response)

Reviewer #2: All comments have been addressed

Reviewer #3: All comments have been addressed

2. Is the manuscript technically sound, and do the data support the conclusions?

Reviewer #1: Partly

Reviewer #2: Yes

Reviewer #3: Yes

3. Has the statistical analysis been performed appropriately and rigorously? 

Reviewer #1: I Don't Know

Reviewer #2: Yes

Reviewer #3: Yes

4. Have the authors made all data underlying the findings in their manuscript fully available?

Reviewer #1: Yes

Reviewer #2: Yes

Reviewer #3: Yes

5. Is the manuscript presented in an intelligible fashion and written in standard English?

Reviewer #1: Yes

Reviewer #2: Yes

Reviewer #3: Yes

6. Review Comments to the Author

Reviewer #1: I feel that the authors made many positive changes in this manuscript. However, I still have some lingering questions and concerns, which I detail below.

The authors seem to be separating decision making and affordances, and I am wondering how different these two concepts are. While I am very familiar with the literature on affordances, I know less about the decision-making in sports literature. However, from the examples that the authors provide, they seem fairly similar. For example, the findings that more experienced surfers are better at determining whether or not a wave is surfable sounds almost identical to the findings in the literature on affordance perception in experts (e.g. Correia, 2012 or Weast, Shockley, & Riley, 2011). I think it would be helpful for the authors to clarify how decision making and affordances are similar or different (i.e. are there meaningful distinctions between these two terms or are two fields of researchers calling the same things by two different names).

In addition, throughout the authors make the argument that affordances were made “more demanding” (Page 7-8 lines 192-193 and elsewhere) and I think this is a bit of a tricky argument to make. In the view of Gibson, affordances are either perceived or they aren’t (i.e. an actor either perceives a possibility for action or they don’t), so affordance perception can’t really be made more demanding. For example, someone with a narrow body frame may perceive an aperture as passable, whereas someone with a wider body frame would not. It is not as if the affordance is more demanding for the person with a wider body frame, it just isn’t perceived. In an athletic context, a more advanced climber may perceive a very small hand hold as graspable, whereas a less experienced climber would not. The affordance is not more demanding for the less experienced climber, they just don’t see that hold as graspable. I think the modifications in experiment 2 added more opportunity for misperception of affordances, but did not make affordance perception more demanding. I think a clearer distinction between decision making and affordances will help with this. One way to frame things would be to say that perception of affordances is an initial process, and then based on the actions that a climber thinks they can complete they have to then craft those potential actions into a motor plan.

I appreciated the extra explanation in the methods and results, but I am still a little stuck on the idea of a “best-solution”. There are so many variables that could impact what is someone’s own “best solution”, and I still think the use of a general “best solution” is somewhat problematic. If I am climbing, my performance does not depends on how I climb the route, but instead if I am able to make it to the top and how quickly I do so.

I still found the results section somewhat cumbersome to read. I think a clear one sentence description of the take away from each experiment immediately following the results section would be useful. I think given the mixed findings across groups; the results sections can be hard to follow. Thus, the authors may have to put in some extra effort to make their findings clear and easy to follow.

Minor Comments:

Page 3 line 74, what does “more appropriate” mean?

Page 6 line 140 “could observe” should be “observed”, same on line 147 and 151, as well as in a few other places throughout the paper.

Reviewer #2: Thank you for addressing my comments. I think the paper has been improved as a result and the paper makes an important contribution to the literature.

Reviewer #3: Authors meaningfully improved the manuscript. In the current status, the article can be accepted for publication.

7. PLOS authors have the option to publish the peer review history of their article (what does this mean?). If published, this will include your full peer review and any attached files.

Reviewer #1: No

Reviewer #2: No

Reviewer #3: No

---

## [Author Response · Author response to Decision Letter 1]

18 Mar 2021

After having profoundly re-read the manuscript, we agree on the comments made by the reviewer #1 and we would like to thank him for the useful suggestions he made. We uploaded a rebuttal letter that responds to each concerns of reviewer #1. We hope that our revised manuscript now fully meets PLOS ONE´s publication criteria.

---

## [Decision Letter · Decision Letter 2]

13 Apr 2021

Effects of Decision-making on Indoor Bouldering Performances: A multi-experimental Study Approach

PONE-D-20-24969R2

Dear Dr. Medernach,

We’re pleased to inform you that your manuscript has been judged scientifically suitable for publication and will be formally accepted for publication once it meets all outstanding technical requirements.

Kind regards,

Luca Paolo Ardigò, Ph.D.

Academic Editor

PLOS ONE

Additional Editor Comments (optional):

Please, address final Reviewer 1's issue while checking proof. Congratulations for the good job.

Reviewers' comments:

Reviewer's Responses to Questions

**Comments to the Author**

1. If the authors have adequately addressed your comments raised in a previous round of review and you feel that this manuscript is now acceptable for publication, you may indicate that here to bypass the “Comments to the Author” section, enter your conflict of interest statement in the “Confidential to Editor” section, and submit your "Accept" recommendation.

Reviewer #1: All comments have been addressed

2. Is the manuscript technically sound, and do the data support the conclusions?

Reviewer #1: (No Response)

3. Has the statistical analysis been performed appropriately and rigorously? 

Reviewer #1: (No Response)

4. Have the authors made all data underlying the findings in their manuscript fully available?

Reviewer #1: (No Response)

5. Is the manuscript presented in an intelligible fashion and written in standard English?

Reviewer #1: (No Response)

6. Review Comments to the Author

Reviewer #1: I think my concerns were adequately adressed. My only minor comment is that on line 134 of the revision the parenthetical description of action capabilities is not quite right since the capability for action is an interaction between the environment and the observer. Maybe action possiblities would be a better word here?

7. PLOS authors have the option to publish the peer review history of their article (what does this mean?). If published, this will include your full peer review and any attached files.

Reviewer #1: No

---

## [Editor Report · Acceptance letter]

4 May 2021

PONE-D-20-24969R2 

Effects of decision-making on indoor bouldering performances: a multi-experimental study approach 

Dear Dr. Medernach:

I'm pleased to inform you that your manuscript has been deemed suitable for publication in PLOS ONE. Congratulations! Your manuscript is now with our production department. 

Kind regards, 

on behalf of

Dr. Luca Paolo Ardigò 

Academic Editor

PLOS ONE